# Rapid incorporation of Favipiravir by the fast and permissive viral RNA polymerase complex results in SARS-CoV-2 lethal mutagenesis

Ashleigh Shannon [1], Barbara Selisko [1], Nhung-Thi-Tuyet Le[1], Johanna Huchting [2], Franck Touret [3], Géraldine Piorkowski[3], Véronique Fattorini [1], François Ferron [1], Etienne Decroly [1], Chris Meier[2], Bruno Coutard[3], Olve Peersen [4✉] & Bruno Canard[1✉]

The ongoing Corona Virus Disease 2019 (COVID-19) pandemic, caused by severe acute respiratory syndrome coronavirus-2 (SARS-CoV-2), has emphasized the urgent need for antiviral therapeutics. The viral RNA-dependent-RNA-polymerase (RdRp) is a promising target with polymerase inhibitors successfully used for the treatment of several viral diseases. We demonstrate here that Favipiravir predominantly exerts an antiviral effect through lethal mutagenesis. The SARS-CoV RdRp complex is at least 10-fold more active than any other viral RdRp known. It possesses both unusually high nucleotide incorporation rates and high-error rates allowing facile insertion of Favipiravir into viral RNA, provoking C-to-U and G-to-A transitions in the already low cytosine content SARS-CoV-2 genome. The coronavirus RdRp complex represents an Achilles heel for SARS-CoV, supporting nucleoside analogues as promising candidates for the treatment of COVID-19.

[1] Architecture et Fonction des Macromolécules Biologiques, CNRS and Aix-Marseille Université, UMR 7257, 13009 Marseille, France. [2] Faculty of Sciences, Department of Chemistry, Organic Chemistry, University of Hamburg, Martin-Luther-King-Platz 6, D-20146 Hamburg, Germany. [3] Unité des Virus Émergents (UVE: Aix-Marseille Univ - IRD 190 - Inserm 1207 - IHU Méditerranée Infection), Marseille, France. [4] Department of Biochemistry & Molecular Biology, Colorado State University, Fort Collins, CO 80523-1870, USA. ✉email: Olve.Peersen@colostate.edu; bruno.canard@afmb.univ-mrs.fr

Coronaviruses (CoV) are large genome, positive-strand RNA viruses of the order *Nidovirales* that have recently attracted global attention due to the ongoing COVID-19 pandemic. Despite significant efforts to control its spread, SARS-CoV-2 has caused substantial health and economic burden, emphasising the immediate need for antiviral treatments. As with all positive strand RNA viruses, an RdRp lies at the core of the viral replication machinery and for CoVs this is the nsp12 protein. The pivotal role of nsp12 in the viral life-cycle, lack of host homologues and high level of sequence and structural conservation makes it an optimal target for therapeutics. However, there has been remarkably little biochemical characterisation of nsp12 and a lack of fundamental data to guide the design of antiviral therapeutics and study their mechanism of action (MoA). A promising class of RdRp inhibitors are nucleoside analogues (NAs), small molecule drugs that are metabolised intracellularly into their active ribonucleoside 5′-triphosphate (RTP) forms and incorporated into the nascent viral RNA by error-prone viral RdRps. This can disrupt RNA synthesis directly via chain termination, or can lead to the accumulation of deleterious mutations in the viral genome. For CoVs, the situation is complicated by the post-replicative repair capacity provided by the nsp14 exonuclease (ExoN) that is essential for maintaining the integrity of their large ~30 kb genomes[1–3]. Nsp14 has been shown to remove certain NAs after insertion by the RdRp into the nascent RNA, thus reducing their antiviral effects[4–6]. Despite this, several NAs currently being used for the treatment of other viral infections have been identified as potential anti-CoV candidates[7–9]. Among these is the purine base analogue T-705 (Favipiravir and Avigan) that has broad-spectrum activity against a number of RNA viruses and is currently licensed in Japan for use in the treatment of influenza virus[10]. Clinical trials are currently ongoing in China, Italy, and the UK for the treatment of COVID-19, although its precise MoA against CoVs has not been shown.

Here we show that a recombinant SARS-CoV RNA polymerase complex has an unusually high polymerisation rate and a very low nucleotide insertion fidelity. This enzyme readily incorporates T-705-ribose-5′-phosphate into viral RNA in vitro, and cell culture based infectious virus studies show an increase in mutations in the presence of Favipiravir. These results indicate Favipiravir can have antiviral effects through alteration of the SARS-CoV-2 genome even in the presence of an active ExoN.

## Results

**T-705 inhibits SARS-CoV-2 through lethal mutagenesis**. We infected Vero cells with CoV-SARS-2 in the presence or absence of 500 μM T-705 (Supplementary Fig. 1a, b) and performed deep sequencing of viral RNA. A 3-fold ($P < 0.001$) increase in total mutation frequencies is observed in viral populations grown in the presence of the drug as compared to the no-drug samples (Fig. 1). Similar to previous findings with influenza[11], Coxsackie B3[5] and ebola[12] viruses, a 12-fold increase in G-to-A and C-to-U transition mutations is observed, consistent with T-705 acting predominantly as a guanosine analogue. The increase in the diversity of the virus variant population suggests that once incorporated into viral RNA, T-705 is acting as a mutagen capable of escaping the CoV repair machinery. Interestingly, the SARS-CoV-2 genome has an already low cytosine content of ~17.6% and T-705 treatment may therefore place additional pressure on its nucleotide content. Associated with this increase in mutation frequency, T-705 has an antiviral effect on SARS-CoV-2, as illustrated by a reduction in virus-induced cytopathic effect, viral RNA copy number and infectious particle yield. Altogether these observations show that the mutagenic effect induced by T-705 is, at least in part, responsible for the inhibition of the replication.

**The highly active SARS-CoV polymerase shows distinct processivity modes**. To determine the efficacy and MoA of T-705 against SARS-CoV we first characterised nsp12 primer-dependent activity using traditional annealed primer-template (PT) and self-priming hairpin (HP) RNAs that may confer additional stability on the elongation complex (Supplementary Fig. 1c). Consistent with prior findings, nsp12 alone is essentially inactive[13,14] and RNA synthesis requires the presence nsp7 and 8 cofactors whose stimulatory effect is enhanced by linking them as a nsp7L8 fusion protein[6,15]. Structures of nsp12 show a four-component complex with a nsp7/nsp8 heterodimer and an additional nsp8 monomer[16,17] and accordingly we found that addition of supplementary nsp8 to the nsp12:nsp7L8 complex further increases activity (Supplementary Fig. 2). The resulting nsp12:7L8:8 complex is highly active on both PT and HP RNAs, with reactions containing 0.2 μM of each substrate and 1 μM nsp12 showing comparable initiation rates, with a rapid burst phase resulting in >50% primer consumption followed by remaining primer use over a period of a few minutes (Fig. 2a, b and Supplementary Fig. 3). Interestingly however, the apparent processivity for the two substrates differs substantially. When provided with an annealed PT pair, intermediate products account for ~60% of the total lane intensity across all timepoints suggesting distributive polymerase activity (Fig. 2c and Supplementary Fig. 4). In contrast, extension of a HP substrate has few intermediate products, indicating a more processive elongation mode. This pattern is consistently observed across RNA substrates of different lengths, showing that the distributive PT mode does not convert into a processive state within a 30-nucleotide long template. Interestingly, a recently resolved structure shows that the two copies of nsp8 form long helical 'sliding pole' extensions that contact the duplex RNA product ~30 basepairs from the active site[18]. In light of this, we believe the lower processivity of PT may indicate a less stable elongation complex for RNA duplexes that are too short to form these contacts. On the other hand, short hairpin substrates are inherently more stably folded and are immune to complete strand separation. This added structural rigidity likely explains the more processive replication on these substrates. Notably, the *Nidovirales* RNA replication/transcription scheme involves precise recombination-like events to generate subgenomic RNAs through a discontinuous mechanism along the ~30,000-nt genome[19]. The different processivities we observe may be connected to these peculiar RNA synthesis events. Differences in RNA secondary structures may significantly alter complex stability, subsequently controlling the dissociation and re-association at specific regions of the RNA template.

**The SARS-CoV polymerase readily incorporates T-705/T-1105 as purine analogues**. We sought to determine the extent to which the RdRp complex could incorporate the NA inhibitors T-705 and T-1105 (a non-fluorinated T-705-related analogue, Fig. 3a) into RNA. T-1105 shows improved potency against influenza virus in certain cell lines and is reportedly more stable than T-705 in the RTP form used in enzyme assays[20–22]. The MoA for these compounds is currently controversial. T-705 has been shown to act through lethal mutagenesis for several viruses, predominantly by competing with guanosine to cause transition mutations[11,23–26]. However, two separate studies support an antiviral effect mediated by chain termination, with incorporation of either a single or two consecutive T-705 molecules blocking further extension by influenza polymerase[27,28].

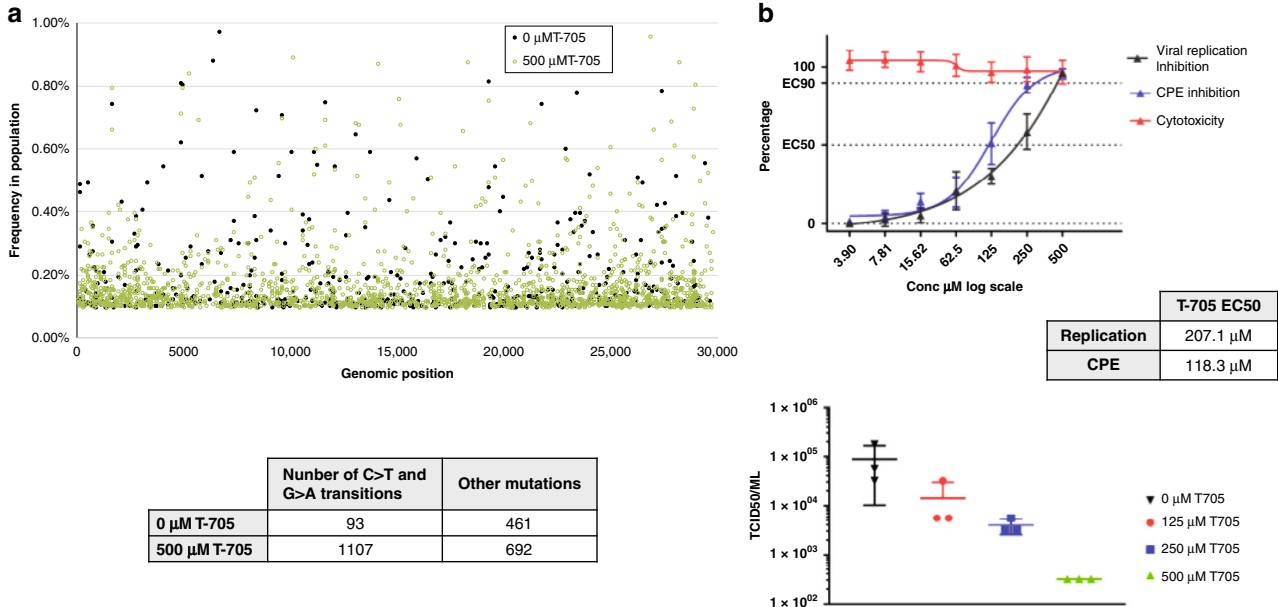

**Fig. 1 Antiviral effects of T-705 on SARS-CoV-2. a** In vitro effects of T-705 on SARS-CoV-2. Distribution of the mutations along the SARS-CoV-2 genome and number of mutations observed in presence or absence of T-705. A 3-fold increase in the presence of the drug is observed ($P < 0.001$, Pearson's $\chi^2$ test with Yates' continuity correction). **b** Quantification of the antiviral effect of T-705 by genome copy number, virus-mediated CPE and number of infectious particles. Mean ± standard deviation (SD) shown ($n = 3$). Source data are provided as a Source Data file.

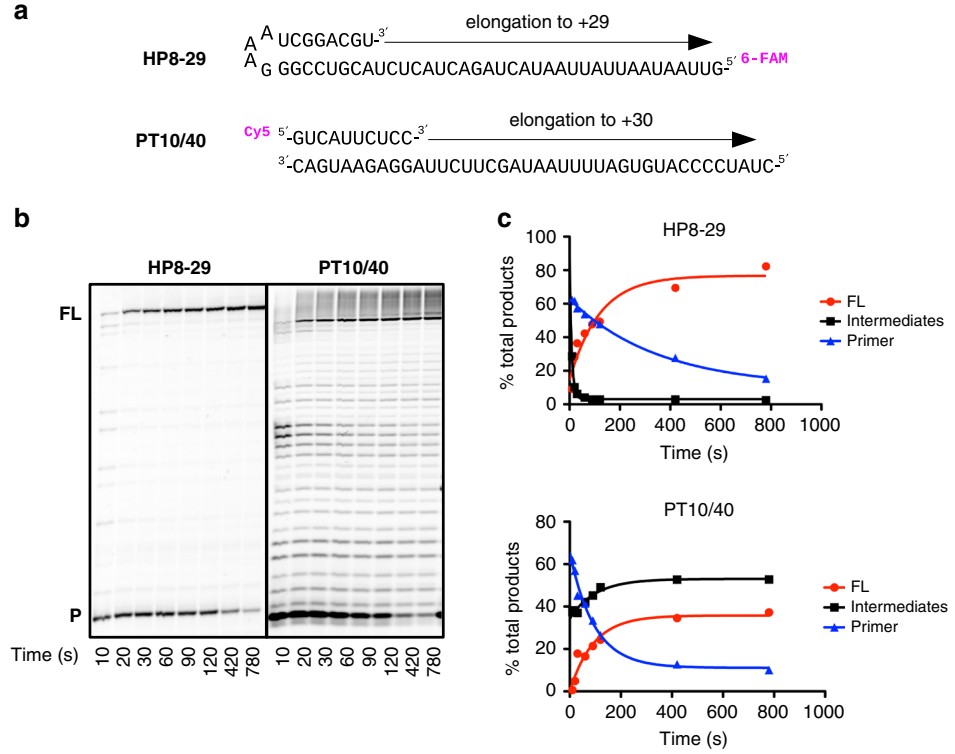

**Fig. 2 Polymerisation modes of SARS-CoV nsp12. a, b** Steady-state data showing extension of hairpin (HP) and annealed primer-template (PT) RNA substrates by the nsp12:7L8:8 complex. **c** Fraction of primer (P, blue), intermediate (black) and full-length (FL, red) species over time, as analysed from gel shown in panel **b**. Additional extension products beyond the full-length observed on the PT substrate are attributed to partially-denatured forms of the full-length product and were included in the analysis of the FL fraction. Source data are provided as a Source Data file.

For the nsp12 complex, omission of ATP and/or GTP from elongation reactions results in rapid incorporation of T-705 and T-1105 at multiple sites with both substrates (Fig. 3b, c). In reactions with 10 μM T-1105-RTP, multiple incorporation events are seen within 10 s, while T-705-RTP is less efficient, potentially attributable to the higher lability of this compound. Neither compound is incorporated in the place of cytosine or uracil, clearly showing they function as purine analogues (Supplementary Fig. 5). For the more processive HP complex, efficient incorporation and elongation occurs opposite both uracil and

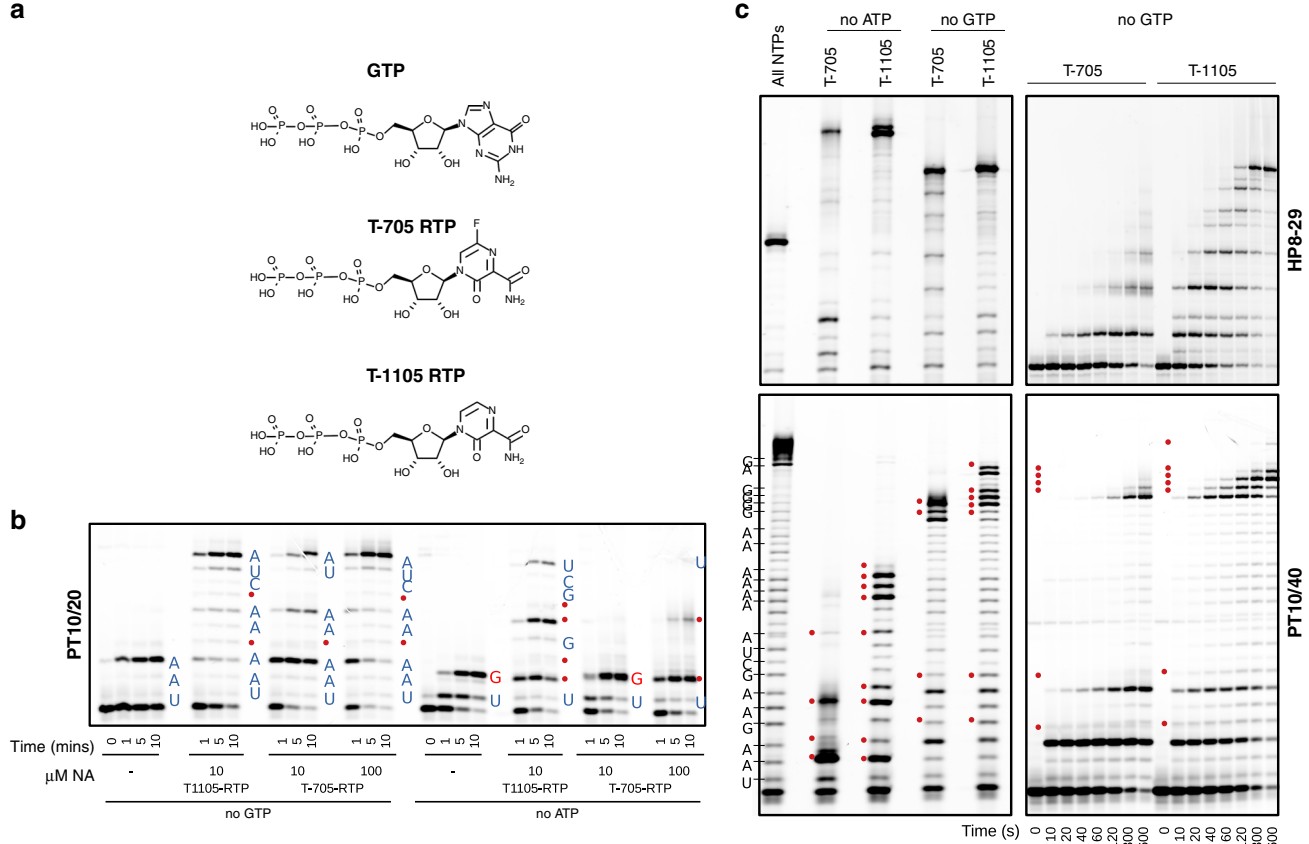

**Fig. 3 Incorporation of nucleoside analogues T-705 and T-1105. a** Structures of GTP, T-705 and T-1105 in their ribonucleotide triphosphate forms.
**b** Elongation of the PT10/20 substrate in the presence of various analogue concentrations and the absence of ATP or GTP. Red dots show analogue incorporation sites at the omitted nucleotide positions, incorporation of correct nucleotides are indicated by blue letters and GTP:U mismatches by red letters. **c** Left panel shows products obtained with 100 µM of each analogue after 5 min. Right panel shows time-course in the presence of 10 µM of each analogue and absence of GTP. Site assignments for the hairpin substrate are not shown due to drastic differences in RNA migration attributed to changes in residual RNA structure following analogue incorporation. Full time-course series at multiple analogue concentrations are shown in Supplementary Fig. 5b. Source data are provided as a Source Data file.

cytosine, resulting in rapid accumulation of full-length products, advocating for lethal mutagenesis as the MoA for these compounds (Fig. 3c). Interestingly, the PT substrate reveals a striking difference in the MoA depending on whether the analogues are incorporated in place of guanine or adenine. Opposite uracil, both are rapidly incorporated but further extension is slow and inefficient. Furthermore, synthesis is seen to stop following multiple consecutive incorporations (Fig. 3c), suggesting these analogues may somewhat destabilise the complex and promote template dissociation, particularly for the less processive PT substrate. In contrast, opposite cytosine there is a stall before each analogue incorporation step, but elongation past the analogue is rapid, even at consecutive sites. More full-length product is observed in these no-GTP experiments, suggesting both compounds are more efficient as guanosine analogues. A similar observation has been made for the poliovirus RdRp that efficiently incorporated and bypassed T-1105 opposite cytosine but was prone to pausing opposite uridine. These pause events were attributed to RdRp backtracking, which was therefore assigned the primary cause of the inhibitory activity[29]. Our results indicate that for nsp12, the T-705/T-1105 MoA is dictated by the structural and functional properties of both the polymerase and the RNA. While the presence of abortive synthesis products suggest that chain termination may contribute to the antiviral effect, this notably only occurred following several consecutive analogue incorporations, a situation that is relatively unlikely

during viral replication. Based on the speed and frequency of analogue incorporation under multiple sequence contexts, we conclude that T-705/T-1105 predominantly act as mutagens, consistent with our infectious virus results.

The reactions performed without ATP also showed detectable amounts of GTP:U mismatch products, allowing us to make a direct comparison of the T-1105 incorporation rate with that of a natural GTP:U mismatch (Fig. 4). Reactions using 50 µM GTP and 1 µM T-1105-RTP show ~5-fold more T-1105:U produced relative to GTP:U. Considering the concentration difference, T-1105 incorporation may be as much as ~250-fold more efficient than the native GTP mismatch, the most common naturally occurring transition mutation. Such high levels of incorporation may exceed the capacity the CoV error-correcting mechanisms, consistent with our infectious virus data, although it remains to be determined if nsp14 can excise T-705/T-1105, as is the case for ribavirin and 5-fluorouracil[4,6,30].

**The SARS-CoV polymerase is the fastest viral RdRp known.** We carried out pre-steady state rapid-quench experiments to further understand the molecular basis of nsp12 elongation rates and fidelity. We initially attempted to form stalled elongation complexes, as is commonly done for the structurally related picornaviral RdRps[31–33], but nsp12 complexes showed half-lives of only ~1 min across a range of conditions and cofactor stoichiometries (Supplementary Fig. 6). The significant rapid burst

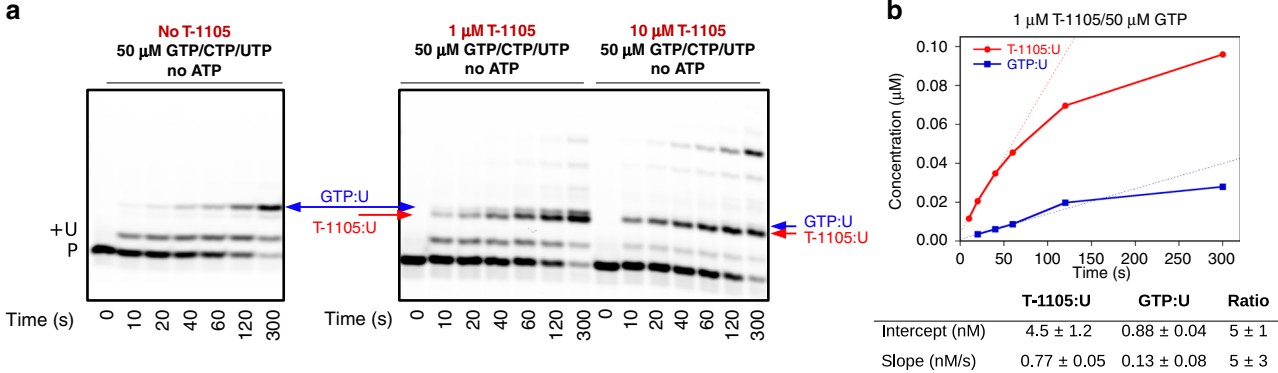

**Fig. 4 Comparison of T-1105-RTP:U and natural GTP:U mismatch incorporation levels. a** Elongation reactions with the PT substrate done in the absence of ATP with 50 μM each GTP, UTP and CTP show rapid addition of the first uracil followed by slow misincorporation of a GTP:U mismatch. In the presence of 1 μM T-1105-RTP (with 50 μM each GTP, UTP and CTP) the analogue incorporation is on a similar timescale as the native GTP:U mismatch. **b** Analysis of incorporation levels show a burst phase followed by linear product buildup over a 60-s timeframe that is consistent with the measured lifetime of the nsp12-7L8-8 elongation complex. Both the burst amplitude and linear rate indicate that 1 μM T-1105-RTP is incorporated ~5-fold faster than the natural GTP:U mismatch at 50 μM GTP. Quantitation reflects slopes and intercepts with standard errors obtained from a linear curve fit to the initial rate data (0–60 s). Expanded concentration series shown in Supplementary Fig. 5b, bottom panel. Source data are provided as a Source Data file.

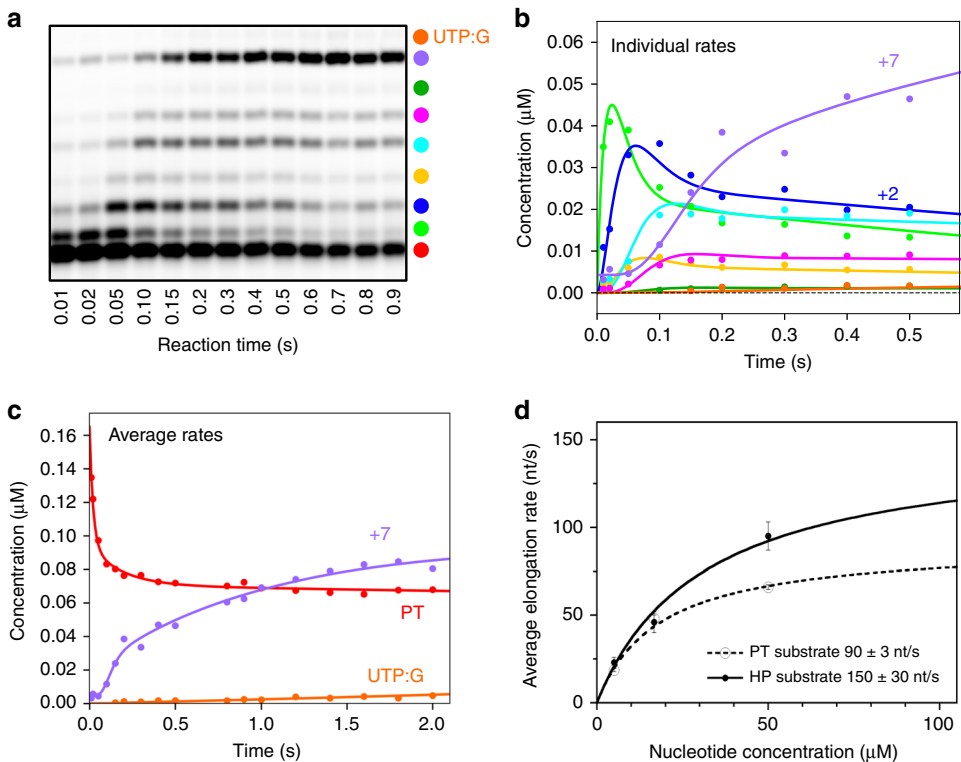

**Fig. 5 Pre steady-state elongation by nsp12-7L8-8 complex. a** EDTA quench-flow data showing rapid elongation to +7 products on the PT10/20 substrate in the absence of CTP. **b, c** Fitting of the quench-flow data to models with discrete rates for each elongation step (**b**, colour key in **a**) or with a single average incorporation rate for each step **c**. **d** Nucleotide concentration dependence yielding maximal elongation rates of $150 \pm 30 \, s^{-1}$ on the HP substrate and $91 \pm 4 \, s^{-1}$ on the PT substrate based on error weighted fits to the observed average elongation rates. Experiments were run at three NTP concentrations with 20 timepoints per experiment, including repeats of 100 ms timepoints ($n = 3$). Supplementary Table 1 lists rates ± standard error (S.E) for both an overall average rate and for each individual incorporation step. Source data are provided as a Source Data file.

phase, however, allowed characterisation of elongation rates under single-turnover conditions using millisecond timescale EDTA quench-flow (Fig. 5 and Supplementary Fig. 7). Experiments were performed at three NTP concentrations based on estimated steady-state $K_d$ values of 3–15 μM. CTP was omitted to allow elongation by only 7 and 8 nucleotides on the PT and HP substrates, respectively, while avoiding end-effects that can slow elongation rates. At 16.7 μM NTP we observe multiple

incorporation steps within a mere 10–20 ms and formation of the +7/8 products by 100 ms (Fig. 5). Analysis of the data using a minimal model of identical nucleotide incorporation steps yields elongation rates of $66 \pm 1.3$ and $95 \pm 8 \, s^{-1}$ for the PT and HP substrates at 50 μM NTP at 22 °C (Supplementary Table 1). Fitting the rate concentration dependence yields maximal elongation rates of $90 \pm 4 \, s^{-1}$ on annealed primer-templates and $150 \pm 30 \, s^{-1}$ on hairpin templates (Fig. 5d).

These data reveal that the SARS-CoV nsp12 is the fastest viral RdRp known, with rates significantly faster than the 5–20 s$^{-1}$ observed for picornaviral polymerases at room temperature[33–35] and 4–18 s$^{-1}$ for hepatitis C and dengue virus polymerases at 30 and 37 °C[36,37]. Based on its structure, nsp12 is expected to use the palm domain based active site closure mechanism that is unique to viral RdRps[32] and which exhibits a 3–5-fold rate increase between 22 °C and 37 °C[34,35], suggesting nsp12 can elongate at 600–700 s$^{-1}$ at physiological temperatures.

**The SARS-CoV polymerase exhibits low nucleotide insertion fidelity.** Such a fast viral RdRp is consistent with the need to rapidly replicate ~30,000-nt-long RNA genomes, but raises questions as to how fidelity of nucleotide incorporation is impacted. In our data, we consistently observe nucleotide misincorporations and subsequent elongation on templates where nsp12 should stall due to the lack of CTP (Supplementary Fig. 2b, 3). In contrast, RdRps that form highly stable elongation complexes show limited or no read-through products under comparable conditions[31,36,38]. The efficiency of the nsp12 bypass is UTP concentration dependent (Supplementary Fig. 3), indicative of uracil misincorporation opposite a templating guanosine (UTP:G). The UTP:G elongation product is also observed in the PT quench-flow data at reaction times as short as 100 ms, yielding a misincorporation rate of ~0.2 s$^{-1}$ (Supplementary Table 1). This is only 400–500-fold lower than the cognate UTP:A rate measured in the same experiment, and more than one order of magnitude less accurate than the generally admitted $10^{-4}$–$10^{-6}$ error rate of viral RdRps[39].

**Structural markers of large Nidovirus RdRp active sites.** The molecular basis for fast and low fidelity replication by nsp12 is not yet known, but a comparison of RdRp structures reveals that a key NTP interaction is absent in CoV enzymes (Fig. 6). Viral RdRps use an electrostatic interaction with an arginine residue in motif F to position the NTP during catalysis[32]. For most positive-strand RNA virus polymerases, this arginine is stabilised by a salt bridge to a glutamic acid residue, also from motif F. Notably, the CoV nsp12 has an alanine in place of this glutamate (A547) and as a result, the arginine (R555) is not rigidly anchored above the active site (Fig. 6a, b)[32,40]. This flexibility could allow catalysis to occur with a relaxed triphosphate positioning, decreasing fidelity

by lowering the requirement for strict Watson-Crick base pairing in the active site. These unique features have likely played a central role in genome expansion and stability by providing a fast RNA synthesis machinery whose inaccuracy is mitigated by the presence of an RNA repair exonuclease. Our data demonstrate that nucleoside analogues are pertinent candidates for the treatment of COVID-19. Favipiravir, with its already defined safety profile and mode of action, may well find a place as an anti-RdRp component in combination therapies targeting coronaviruses.

## Method

**Nucleotide substrates.** T-1105-ribose-TP (RTP), the 5′-triphosphate of the non-fluorinated derivative of T-705-ribose, was synthesised following the iterative route from mono- via di- to triphosphate[22] with minor modifications (Supplementary Fig. 8). To overcome low solubility of T-1105-ribonucleotides in organic solvent, the 2′- and 3′-acetyl groups were kept until the final TP-synthesis and were then cleaved by treatment with base. This reaction sequence resulted in an increased yield of up to 90% (for di-) and 46% (for triphosphate), nearly double of previously reported. T-705-RTP was obtained from Toronto Research Chemicals. Lyophilised aliquots were resuspended in TE buffer pH 7.0 and the stability was verified before each experiment by measuring absorbance at 372 nm and 350 nm for T-705-RTP and T-1105-RTP respectively. Stability was additionally checked through HPLC analysis of both compounds fresh and after dilution in the assay reaction buffer and left at room temperature for 20 min. No loss of the active triphosphate form was observed. Other NTPs were purchased from GE Healthcare.

**Synthetic oligonucleotides.** Primer-template (PT) pairs were purchased from Biomers (HPLC grade). RNA oligonucleotides ST20 (20-mer) and LS1 (40-mer), corresponding to the 3′-end of the SARS-CoV genome (excluding the poly A tail) were used as templates, annealed to a 5′ cy5 labelled SP10 primer (10-mer) corresponding to the 5′-end of the anti-genome. For simplicity, these were named PT10/20 and PT10/40 throughout the manuscript. Annealing was performed by denaturing primer:template pairs (molar ratio of 1:1.5) in 110 mM KCl at 70 °C for 10 min, then cooling slowly to room temperature over several hours. Hairpin RNAs were synthesised by Integrated DNA Technologies (Coralville, IA), resuspended at 100 μM concentration in 50 mM NaCl, 5 mM MgCl$_2$, 10 mM Tris (pH 8.0) and heated to 95 °C for 15 min before snap cooling on ice.

**Expression and purification of SARS-CoV proteins.** All SARS-CoV proteins used in this study were expressed in *Escherichia coli* (*E. coli*) under the control of T5 promoters (primers used for cloning shown in Supplementary Table 2). Cofactors nsp7L8 and nsp8 alone were expressed from pQE30 vectors with C-terminal and N-terminal hexa-histidine tags respectively. TEV cleavage site sequences were included for His-tag removal following expression. The nsp7L8 fusion protein was generated by inserting a GSGSGS linker between nsp7- and nsp8-coding sequences. Cofactors were expressed in NEB Express C2523 (New England Biolabs) cells carrying the pRare2LacI (Novagen) plasmid in the presence

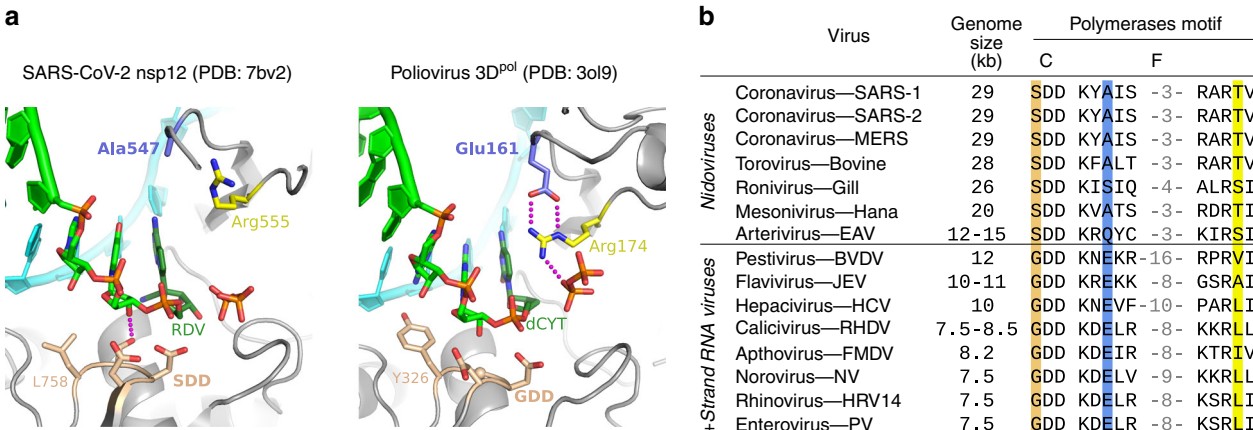

**Fig. 6 Comparison of viral polymerase active site structures and sequences. a** Analogous post-catalysis, pre-translocation structures of CoV nsp12 and poliovirus 3D$^{pol}$ after incorporation of remdesivir (RDV) and deoxycytosine (dCYT), respectively. The CoV polymerases have replaced a motif F glutamate residue (3D$^{pol}$ Glu161) with an alanine (nsp12 Ala547), removing a highly conserved interaction that positions the motif F arginine for interactions with the NTP and pyrophosphate. **b** Alignment of representative viral RdRp sequences showing the large genome nidoviruses have a SDD instead of GDD sequence in the palm domain motif C (tan) and alanine, serine or glutamine in place of the aforementioned glutamate (blue) found in other positive strand RNA viruses. The NTP-interacting arginine (yellow) is conserved, but the overall length of the motif F loop is shorter in the nidoviruses (numbers reflect omitted residues).

of Ampicillin (100 μM/mL) and Chloramphenicol (17 μg/mL). Protein expression was induced with 100 μM IPTG once the $OD_{600} = 0.5$–0.6, and expressed overnight at 17 °C. Cells were lysed by sonication in a lysis buffer containing 50 mM Tris-HCl pH 8, 300 mM NaCl, 10 mM Imidazole, supplemented with 20 mM MgSO₄, 0.25 mg/mL Lysozyme, 10 μg/mL DNase and 1 mM PMSF. Protein was purified first through affinity chromatography by batch binding to HisPur Cobalt resin (Thermo Scientific) followed by elution with lysis buffer supplemented with 250 mM imidazole. Eluted protein was concentrated and dialysed overnight in the presence of histidine labelled TEV protease (1:10 w/w ratio to TEV:protein) for removal of the protein tag. Cleaved protein was void volume purified through a second cobalt column and subjected to size exclusion chromatography (GE, Superdex S200) in gel filtration buffer (25 mM HEPES pH 8, 150 mM NaCl, 5 mM MgCl₂ and 5 mM TCEP). Concentrated aliquots of protein were flash-frozen in liquid nitrogen and stored at −80 °C. A synthetic, codon-optimised SARS-CoV nsp12 gene (Supplementary Table 3) bearing C-terminal 8xHis-tag preceded by a TEV protease cleavage site was expressed from a pJ404 vector (DNA 2.0) in *E. coli* strain BL21/pG-Tf2 (Takara). Cells were grown at 37 °C in the presence of Ampicillin and Chloramphenicol until $OD_{600}$ reached 2. Cultures were induced with 250 μM IPTG and protein expressed at 17 °C overnight. Purification was performed as above in lysis buffer supplemented with 1% CHAPS. Two additional wash steps were performed prior to elution, with buffer supplemented with 20 mM imidazole and 50 mM arginine for the first and second washes respectively. Polymerase was eluted using lysis buffer with 500 mM imidazole and concentrated protein was purified through gel filtration chromatography (GE, Superdex S200) in the same buffer as for nsp7L8. Collected fractions were concentrated and supplemented with 50% glycerol final concentration and stored at −20 °C.

**Steady-state elongation complex reactions.** Nsp12, nsp7L8 and nsp8 were mixed just prior to each experiment at a 1:3:3 molar ratio (unless otherwise stated) and preincubated on ice for 10 mins. The protein complex was subsequently mixed with the RNA pre-mix (20 mM HEPES pH 7.5, 50 mM NaCl and 5 mM MgCl₂) containing either a single RNA substrate or both HP and PT RNAs at equimolar ratios. Reactions were initiated with 50 μM (final concentration) of all four NTPs, or without CTP for partial elongation reactions. Final reaction concentrations were 1 μM nsp12, 0.2 μM each RNA. Reactions were quenched at indicated timepoints with 5X volume of FBD stop solution (formamide, 10 mM EDTA). To verify that activity was not due to either nsp7L8 or nsp8, which have been shown to harbour primase-like non-canonical RdRp activities[41–44] control assays were run in the absence of nsp12. The NTP $K_d$ was estimated using the same conditions, but with final equimolar concentration of ATP, GTP and UTP ranging from 0.78 to 100 μM (2-fold dilution series). The time course of product formation was fit to single exponential equation for each concentration of NTP to give the observed rate constant ($k_{obs}$). Observed rates were subsequently plotted against NTP concentration, and the data was fit via hyperbolic regression to give the equilibrium dissociation constant ($K_d$) and the maximum rate constant for incorporation of NTPs ($k_{pol}$).

**Formation of stalled elongation complex.** Attempts to form a stalled elongation complex were performed with HP8-22 RNA using a constant nsp12 concentration of 0.8 μM with varied nsp12:nsp7L8:nsp8 ratios (Supplementary Fig. 6). Protein, HP8-22 RNA (0.2 μM) and 50 μM final concentrations of GTP and ATP were incubated for 5 min at room temperature to allow formation of a stalled +4-nucleotide elongation complex. Reactions were diluted 1:2 in high salt to prevent RNA rebinding and chased at various timepoints (1–50 min) with 50 μM all NTPs. Chase reactions were quenched after 30 s in FBD. Stability half-life was calculated from the ratio of full-length (FL) product produced relative to the total amount of +4-lock + FL product at each timepoint. Half-life was obtained by fitting the data through a single exponential.

**Pre-steady state quenched-flow kinetics.** Experiments were carried out in a Bio-Logic QFM-300 rapid chemical quench-flow apparatus that controls reaction time by flow rate through a 2.5 μL chamber between the reaction mixer and the quencher mixer. The nsp12-nsp7L8-nsp8 complex was assembled by first pre-incubating the proteins on ice for 10 min at a 1:3:3 molar ratio, then adding this to SP10/20 and HP8-29 hairpin RNA in reaction buffer (20 mM HEPES pH 7.5, 5 mM MgCl₂, 50 mM NaCl) and further incubated at room temperature for 15 min prior to beginning the rapid-quench experiments, which took ~10 min to complete a set of 22 different reaction times. Reactions were initiated by mixing equal volumes (18 μl each) of the protein-RNA complex with NTP solutions containing equimolar concentrations of ATP, GTP and UTP, and then quenched with 18 μL of 200 mM EDTA. Final concentrations in the reactions were 1 μM nsp12, 0.2 μM primer-template RNA, 0.2 μM hairpin RNA and 50, 16.7 or 5.6 μM NTP, and the final EDTA concentration after quenching was 67 mM.

**Product analysis.** Quenched reactions were mixed 1:1 with FBD loading dye and heated for 10 min at 70 °C, and cooled on ice for 2 min before analysis on 17–20% polyacrylamide, 7 M Urea TBE gels. Gels were run at a constant 65 W using Sequi-Gen GT Systems from Bio-Rad or vertical electrophoresis systems from CBS Scientific and visualised using an Amersham™ Typhoon™ Biomolecular Imager (GE Healthcare). The intensity of each band was quantified using the ImageQuant

software (GE Healthcare)/Image Gauge (Fuji) and/or using ImageJ as implemented in the Fiji package, with background subtraction. Product yield was determined by dividing the intensity of the product by total intensity of the product + remaining primer and multiplying by the input concentration of RNA. For the rapid quench data, the programme PeakFit (Systat Software) was used to fit gel lane profiles to a set of gaussian peaks and the fractional area contained within each peak was multiplied by the RNA concentration in the experiment to calculate the amount of each elongation species as a function of reaction time.

**Pre-steady state kinetic data analysis.** Rapid quench data were analysed using the programme KinTek Explorer[45] to model the reactions as a series of seven (PT substrate) or eight (HP substrate) irreversible nucleotide addition steps. Data from each NTP concentration series were fit independently (not globally) to obtain observed rates using either a model with (i) a single average rate or (ii) individual rates for each nucleotide addition step (Supplementary Fig. 7). Modelling the remaining intermediate species attributed to low processivity elongation using a formal RNA dissociation step and a rebinding step at rates comparable to non-burst phase primer utilisation was a not successful, suggesting the nsp12-RNA complex exists in some form of inactive state. Low processivity elongation was instead accommodated in the kinetic model by adding a reversible inactivation/reactivation equilibrium step for each elongation product, allowing intermediate products to depart from the immediate processive pathway toward full-length product formation, but then rejoin the pathway for further elongation. The rate constants for these steps were shared among all intermediates species as there was insufficient data to fit them individually.

The single average rate models were fit using only data from primer loss and the full-length product (+7 or +8), but not the intermediate species. Note that for the PT template we observed detectable amounts of additional +8 and +9 bands at the higher 16.7 and 50 μM NTP concentrations and therefore included them in the model. This is presumably due to a UTP:G mismatch to yield the +8 product in the absence of CTP, followed by a cognate ATP:U addition that is slow because it is priming on a mismatched base pair. For the HP substrate, we were unable to definitively identify the gel migration bands for the +6 and +7 species and therefore did not model individual rates for these two elongation events. Data from the HP RNAs are challenging to analyse because the high thermodynamic stability of the folded RNA structure can make it difficult to fully denature the helix during electrophoresis. This leads to a mixture of species that transition from denatured single-stranded RNA for the initial species to more compact and faster migrating duplexes for longer elongation products. We were therefore unable to reliably analyse quench flow data from longer hairpin products.

**NTP-analogue incorporation.** Enzyme mix (20 μM nsp12, 60 nsp7L8 and 60 μM nsp8) in complex buffer (25 mM HEPES pH 7.5, 150 NaCl, 5 mM TCEP and 5 mM MgCl₂) was incubated 10 min on ice and then diluted in reaction buffer (20 mM HEPES pH 7.5, 50 mM NaCl and 5 mM MgCl₂) to 4 μM nsp12 (12 nsp7L8 and 8). The resulting enzyme complex was mixed with an equal volume of 0.8 μM primer/template (PT) with or without 0.8 μM hairpin (HP) carrying Cy5 or 6-FAM fluorescent labels, respectively, at their 5′ ends in reaction buffer, and incubated for 10 min at 25 °C or 30 °C, as given in Figure legends. Reactions were then started with the same volume of 100 μM NTPs (leaving out one or two as given in figure legends) and TE buffer or NTP-analogs (T-1105-RTP or T-705-RTP) in reaction buffer. Final concentrations in the reactions were 1 μM nsp12 (3 μM nsp7L8 and 8), 0.2 μM PT (and 0.2 μM HP), 50 μM NTPs and the given concentrations of NTP-analogs. Samples of 8 μl were taken at given time points and mixed with 40 μl of formamide containing 10 mM EDTA. In all, 10-μl samples were analysed by denaturing PAGE (20% acrylamide, 7 M urea, TBE buffer) and visualised by a fluorescence imager (Amersham Typhoon).

**In vitro infection assays.** Cell line: VeroE6 (ATCC CRL-1586) cells were grown in minimal essential medium (Life Technologies) with 7.5% heat-inactivated fetal calf serum (FCS), at 37 °C with 5% CO₂ with 1% penicillin/streptomycin (PS, 5000 U mL⁻¹ and 5000 μg mL⁻¹ respectively; Life Technologies) and supplemented with 1% non-essential amino acids (Life Technologies).

Virus strain: SARS-CoV-2 strain BavPat1 was obtained from Pr. Drosten through EVA GLOBAL (https://www.european-virus-archive.com/). Virus stocks were prepared using standard methods[46]. All experiments were conducted in a BSL3 laboratory.

Antiviral experiments: for EC50 and CC50 determination, 1 day prior to infection, $5 \times 10^4$ VeroE6 cells were seeded in 100 μL assay medium (containing 2.5% FCS) in 96-well plates. The next day, seven 2-fold serial dilutions of T-705 (500 μM to 3.9 μM in triplicate) were added to the cells (25 μL/well, in assay medium). Four virus control wells were supplemented with 25 μL of assay medium. After 15 min, 25 μL of a virus mix diluted in medium was added to the wells. The amount of virus working stock used was calibrated prior to the assay based on replication kinetics so that the replication growth is still in the exponential growth phase for the readout[47,48]. Four cell control wells (i.e. with no virus) were supplemented with 50 μL of assay medium. Plates were incubated for 2 days at 37 °C prior to quantification of the viral genome by real-time RT-PCR. RNA extraction and viral RNA quantification was performed[46]. The 50 and 90% effective

concentrations (EC50, EC90; compound concentration required to inhibit viral RNA replication by 50 and 90%) were determined using logarithmic interpolation[47]. For the evaluation of CC50 (the concentration that reduces the total cell number by 50%), the same culture conditions were set as for the determination of the EC50, without addition of the virus, then cell viability was measured using CellTiter Blue® (Promega). Cell supernatant media were discarded and CellTiter-Blue® reagent (Promega) was added following the manufacturer's instructions. Plates were incubated for 2 h prior recording fluorescence (560/590 nm) with a Tecan Infinite 200Pro machine. From the measured $OD_{590}$, CC50 was determined using logarithmic interpolation. For EC50 determination using CPE inhibition, cells and T-705 were prepared as described above. Eight virus control wells were supplemented with 25 µL of assay medium and eight cell control were supplemented with 50 µL. After 15 min, 25 µL of a virus mix diluted in 2.5% FCS-containing medium was added to the wells at MOI 0.002. Three days after infection, CPE were assessed using CellTiter-Blue® reagent (Promega). For the infectivity test cells and compound were prepared as described above (EC50 determination) with only three 2-fold serial dilutions of T-705 (125 µM to 500 µM, in triplicate). Three virus control wells were supplemented with 25 µL of assay medium. The experiment was conducted as described for the EC50 determination. At day 3 the supernatant was collected and each triplicate was titrated by measuring the 50% tissue culture infectivity dose ($TCID_{50}$); briefly, three replicates were infected with 100 µL of 10-fold serial dilutions of T-705 (125 µM to 500 µM, and incubated for 3 days. CPE was measured by CellTiter-Blue® reagent (Promega) and TCID50 was calculated and expressed as $TCID_{50}/mL$. Antiviral experiments data were analysed using GraphPad Prism 7 software (Graph pad software). Graphical representations were also performed on GraphPad Prism 7 software.

**Sequence analysis.** Eight overlapping amplicons were produced from the extracted viral RNA using the SuperScript *IV* One-Step *RT-PCR System* (Thermo Fisher Scientific) and specific primers (Supplementary Table 4). PCR products were pooled at equimolar proportions. After Qubit quantification using Qubit® dsDNA HS Assay Kit and Qubit 2.0 fluorometer (ThermoFisher Scientific) amplicons were fragmented by sonication in ~200 bp long fragments. Libraries were built adding barcodes for sample identification to the fragmented DNA using AB Library Builder System (ThermoFisher Scientific). To pool the barcoded samples at equimolar ratio a quantification step by real-time PCR using Ion Library TaqMan™ Quantitation Kit (Thermo Fisher Scientific) was realised. An emulsion PCR of the pools and loading on 530 chip was realised using the automated Ion Chef instrument (ThermoFisher). Sequencing was performed on the S5 Ion torrent technology v5.12 (Thermo Fisher Scientific) following manufacturer's instructions. Consensus sequence was obtained after trimming of reads (reads with quality score <0.99, and length <100 pb were removed and the 30 first and 30 last nucleotides were removed from the reads) mapping of the reads on a reference (determined following Blast of De Novo contigs) using CLC genomics workbench software v.20 (Qiagen). A de novo contig was also produced to ensure that the consensus sequence was not affected by the reference sequence. Quasi species with frequency over 0.1% were studied. Sequencing data generated for this study are available (SRA accession number: PRJNA633443; GenBank accession numbers are MT594401 and MT594402). In parallel with the viral genomes, a cloned SARS-CoV-2 DNA fragment (region 7112–11,365 on MT263402 genome) was treated with the same amplification and sequencing procedure to evaluate the mutation frequency induced by the sequencing steps. No sub-population was observed from the cloned DNA implying that sub-populations observed in the virus samples reflect the sequence diversity induced upstream the amplification process.

**Reporting summary.** Further information on research design is available in the Nature Research Reporting Summary linked to this article.

## Data availability

Sequencing data generated for this study are available (SRA accession number: PRJNA633443; GenBank accession numbers are MT594401 and MT594402). Source data are provided with this paper. Other data are available from the corresponding authors upon reasonable request. Source data are provided with this paper.

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

## Acknowledgements
We thank Magali Gilles and Karine Alvarez for excellent technical support, as well as Prs. C. Drosten and F. Drexler for providing the SARS-CoV-2 through EVA-GLOBAL (European Union's Horizon 2020 programme, GA 871029). This work was supported by the Fondation pour la Recherche Médicale (Aide aux équipes), the SCORE project H2020 SC1-PHE-Coronavirus-2020 (grant#101003627) to BCa, REACTing Covid-19 initiative (REsearch and ACTion targeting emerging infectious diseases) with the support of the Ministry of Solidarity and Health and the Ministry of Higher Eductation to BCa, ED and BCo, National Institutes of Health grant AI059130 to OP, and a grant from DZIF (German Center for Infection Research) to J.H. and C.M.

## Author contributions
B.S., N.T.T.L. and V.F. performed experiments and analysed data; J.H. and C.M. performed experiments; F.T., G.P. and B.Co performed experiments and analysed data; F.F./E.D. analysed data; A.S., O.P. and B.Ca designed experiments, performed experiments, analysed data and wrote the manuscript.

## Competing interests
The authors declare no competing interests.
