## [Peer Review File · Nature Communications]

REVIEWERS' COMMENTS:

Reviewer #1 (Remarks to the Author):

The authors nicely replied to our comments and changed the statements in the manuscript accordingly. Maybe as expected, it is now apparent that chain termination does not play a role in the mechanism of action and the main mechanism of Favipiravir is actually increased mutagenesis, consistent with the literature.

I recommend publication of this important work after minor adjustments that I leave to the editor to judge:

The title is not informative and should be changed to better reflect the contents.

The introduced speculation that primer-template substrates are less well extended compared to hairpin substrates because unstructured portions of nsp8 might unwind the RNA duplex is not supported and may be deleted.

Reviewer #2 (Remarks to the Author):

I previously reviewed this manuscript by Shannon et al for Nature. In this revised version, the authors have addressed my comments in sufficient detail by either clarifying the text and/or providing additional data. Any further points I had have been covered by comments to the other reviewers. Overall, I feel that the current data support the claims made in the manuscript and that the manuscript would be a valuable addition to the field, in particular in light of the current pandemic

Rapid incorporation of Favipiravir by the fast and permissive viral RNA polymerase complex results in SARS-CoV-2 lethal mutagenesis

Shannon et al., 2020

Response to Referees

REVIEWERS' COMMENTS:

Reviewer #1 (Remarks to the Author):

The authors nicely replied to our comments and changed the statements in the manuscript accordingly. Maybe as expected, it is now apparent that chain termination does not play a role in the mechanism of action and the main mechanism of Favipiravir is actually increased mutagenesis, consistent with the literature.

I recommend publication of this important work after minor adjustments that I leave to the editor to judge:

The title is not informative and should be changed to better reflect the contents. **Title has been changed**

The introduced speculation that primer-template substrates are less well extended compared to hairpin substrates because unstructured portions of nsp8 might unwind the RNA duplex is not supported and may be deleted. **The statement 'or that can perhaps be strand separated by interactions with unfolded sections of nsp8' has been removed from line 103**

Reviewer #2 (Remarks to the Author):

I previously reviewed this manuscript by Shannon et al for Nature. In this revised version, the authors have addressed my comments in sufficient detail by either clarifying the text and/or providing additional data. Any further points I had have been covered by comments to the other reviewers. Overall, I feel that the current data support the claims made in the manuscript and that the manuscript would be a valuable addition to the field, in particular in light of the current pandemic. **No changes requested**